# BEYOND MODEL-CENTRIC: COLLABORATIVE DATA OPTIMIZATION FOR REUSING AND SHARING

## ABSTRACT

This paper pioneers a *novel data-centric paradigm* to maximize the utility of unlabeled data, tackling a critical question: *How can we enhance the sustainability and efficiency of deep learning training by optimizing the data itself?* We begin by identifying two key limitations in existing model-centric approaches, all rooted in a shared bottleneck: knowledge extracted from data is locked to model parameters, hindering its reusability and scalability. To this end, we propose COOPT, a highly efficient, parallelized framework for collaborative unlabeled data optimization. By distributing unlabeled data and leveraging publicly available task-agnostic prior models, COOPT optimizes raw unlabeled data into knowledge-enriched training sets that are effective, efficient, reusable, and easily shareable. Extensive experiments across diverse datasets and architectures validate these advantages, achieving a 7.9% improvement on ImageNet-1K over BYOL. Notably, COOPT remains effective even when all prior models are significantly weak, substantially accelerating the early stages of training. These results establish data-centric optimization as a promising path toward sustainable and efficient deep learning [1].

## 1 INTRODUCTION

Deep Learning has achieved remarkable success, primarily due to the large-scale datasets (Song et al., 2020; Yang et al., 2023). Despite the abundance of data in the era of big data, a significant portion of them remains unlabeled (Lei & Tao, 2023). The dominant paradigm in the field for exploiting unlabeled data is self-supervised learning (SSL), which is fundamentally *model-centric*: it carefully crafted pretext tasks and loss functions to encode data information into model parameters (Chen et al., 2020a; Grill et al., 2020; Gui et al., 2024).

However, the model-centric nature presents two critical challenges. ***First***, their training protocols are tightly coupled to specific network architectures, severely hindering the transferability and reusability of trained model to other architectures (Wagner et al., 2022; Huang et al., 2023). ***Second***, despite acceleration advances, training over extensive unlabeled datasets still computationally prohibitive (Sun et al., 2024). At the core of these challenges

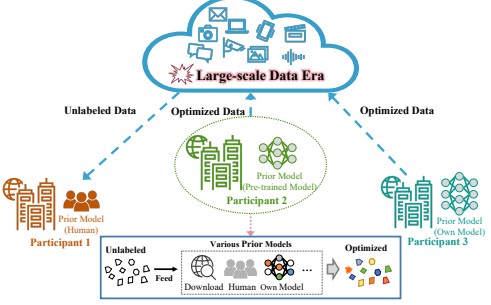

**Figure 1: A Collaborative Data Optimization Framework COOPT.** For large-scale unlabeled data, self-supervised learning results in 😣 low training efficiency. Therefore, we propose COOPT, an 😊 efficient and parallel framework enabling participants to use diverse task-agnostic models, such as pre-trained ResNets, termed *prior models*, for collaborative data optimization.

is a shared bottleneck: knowledge extracted from data is locked in model parameters, restricting its adaptability and preventing efficient reuse across diverse tasks or architectures.

To break free from model-centric paradigm, we propose a data-centric paradigm that directly optimizes the unlabeled data by optimizing targets for samples (detailed in Sec. 3.1), thereby effectively encoding knowledge into the data itself rather than into model parameters. The resulting "optimal data" is agnostic to downstream architectures, accelerates subsequent training by providing richer supervision, and can be reused across multiple tasks without repeated large-scale pretraining.

---

[1] Our code is provided in the Supplementary Materials and will be publicly accessible.

In the meanwhile, scaling this approach to massive unlabeled datasets introduces a significant challenge: a single node faces prohibitive compute and storage demands. To address this, we propose CoOpt, a highly efficient collaborative framework inspired by crowd-sourcing to achieve parallel data optimization. An overview of CoOpt is depicted in Fig. 1, with detailed processes shown in Fig. 3. In CoOpt, the unlabeled dataset is partitioned into disjoint subsets, each processed independently and in parallel by participants equipped with task-agnostic models. These models, referred to as *prior models*, can diverge from publicly available pre-trained models and the participants' local models. Once the targets are optimized, they are aggregated to reconstruct a fully optimized dataset, achieving computational scalability through decentralized workload distribution.

CoOpt offers ***three key advantages***: First, relying solely on task-agnostic prior models, the optimized data can be directly transferred to any downstream architecture, thereby ensuring strong generalizability and reusability (see Sec. 4.2). Second, by distributing non-overlapping data subsets across participants, each node handles only a fraction of the total computational cost, thereby enabling scalable and resource-efficient optimization (see Sec. 4.2). Third, CoOpt is lightweight, incurring negligible overhead while substantially enhancing training efficiency and performance (see Sec. 4.4).

**In summary, our contributions are threefold:**

(a) We propose CoOpt, the *first* data-centric framework for collaboratively optimizing unlabeled data. By leveraging task-agnostic prior models, CoOpt transforms raw unlabeled samples into optimal data, enabling high performance, efficiency, strong generalization, and reusability.

(b) Within CoOpt, we identify a critical issue, *Target Distribution Inconsistency* (Sec. 3.3), and introduce a lightweight target alignment strategy to address it (Sec. 3.4).

(c) We conduct experiments across datasets and models to comprehensively validate the advantages of CoOpt (Sec. 4.2). Further, we provide a detailed analysis of the key factors influencing its effectiveness (Sec. 4.3). Remarkably, we demonstrate that CoOpt remains effective even when all prior models are weak, substantially accelerating the early stages of training.

## 2 RELATED WORK

**Self-Supervised Learning.** It aims to exploit the intrinsic relationships within unlabeled data. For example, InstDisc (Wu et al., 2018) uses instance discrimination as a pretext task. MoCo (He et al., 2020) significantly increases the number of negative samples but uses a simplistic strategy for selecting positive samples. SimCLR (Chen et al., 2020a) highlights the importance of hard positive sample strategies. Notably, BYOL (Grill et al., 2020) discards negative sampling and surpasses the performance of SimCLR (Chen et al., 2020a).

**Model-Centric Perspective: Knowledge Distillation.** Knowledge distillation (Hinton, 2015) leverages teacher-generated soft labels to improve student training efficiency and performance (Dong et al., 2023). A line of knowledge distillation methods utilizes multiple teachers (MKD) (Zhang et al., 2022; Pham et al., 2023) to enhance student learning. They assume that ensemble outputs from multiple teachers enables students to learn more generalized representations. Notably, all teacher models process the same input data.

*Our setting departs fundamentally from knowledge distillation in terms of objective, input data, and teacher models (see Fig. 2).* First, in KD, the distilled knowledge is embedded in student parameters, limiting its reuse across different architectures. In contrast, our objective is to construct a high-quality, optimized dataset that is model-agnostic and reusable, enabling training or evaluation of diverse architectures. Second, rather than feeding all teachers the same inputs, we partition the unlabeled data into disjoint subsets, each optimized by a different prior model. Third, existing KD methods often rely on intricate loss functions (Jiang et al., 2024) or require teacher fine-tuning (Wu et al., 2021), but our framework uses arbitrary pre-trained models without domain-specific adaptation. The optimized data can then be directly reused to train arbitrary downstream models without further modification.

**Data-Centric Perspective: Dataset Distillation.** Dataset distillation (Wang et al., 2018) improves the training efficiency by learning a compact distilled dataset that can achieve comparable performance to the original dataset with less training cost. The majority of methods focus on optimizing images, which can be categorized into three primary approaches (Lei & Tao, 2023): meta-learning frameworks (Wang et al., 2018; Zhou et al., 2022), matching-based methods (Zhao et al., 2020; Zhao & Bilen, 2023; Guo et al., 2024) and decoupling frameworks (Yin et al., 2023; Sun et al., 2024). Notably, current methods predominantly focus on distilling labeled datasets.

## 3 COLLABORATIVE DATA OPTIMIZATION FRAMEWORK COOPT

We begin by formally defining *data optimization* in Sec. 3.1. Subsequently, we provide a detailed description of the proposed COOPT in Sec. 3.2. Furthermore, we identify an inherent challenge within this framework in Sec. 3.3 and present method in Sec. 3.4.

### 3.1 DEFINITION OF DATA OPTIMIZATION

We first revisit current training acceleration techniques, including knowledge distillation (Hinton, 2015) and dataset distillation (Wang et al., 2018). Specifically, we decouple data into samples $D_X$ and targets $D_Y$. Essentially, compared to self-supervised learning methods, these approaches are more efficient. They achieve this by either optimizing the target $D_Y$ of pre-trained models (as in knowledge

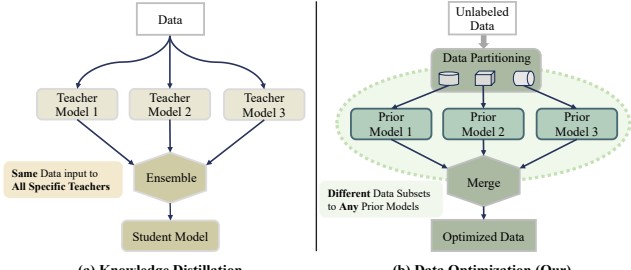

Figure 2: Comparison Between KD and Ours.

distillation) or by jointly optimizing both the input data $D_X$ and target $D_Y$ (as in dataset distillation). Notably, for dataset distillation, optimizing input data $D_X$ is computationally more expensive than optimizing targets $D_Y$ (Bohdal et al., 2020). Furthermore, recent studies Shang et al. (2025); Qin et al. (2024) have indicated that solely optimizing $D_Y$ not only reduces computational overhead but also achieves significant performance gains. These findings have demonstrated that *optimizing $D_Y$ is both necessary and efficient*. We refer to this process as **data optimization**.

Formally, consider a large-scale unlabeled dataset $D = D_X = \{\mathbf{x}_i\}_{i=1}^N$, where $\mathbf{x}_i \in \mathbb{R}^m$ and $N = |D|$, *data optimization* aims to assign targets $D_Y = \{\mathbf{y}_i\}_{i=1}^N$ to construct an optimally labeled dataset $D' = \{(\mathbf{x}_i, \mathbf{y}_i)\}_{i=1}^N$ such that models trained on $D'$ can achieve **higher** performance of those trained on $D$ with **significantly less training costs**. This objective is expressed as

$$\mathbb{E}_{(\mathbf{x},y) \sim P_{\mathcal{T}}}[\ell(\phi_{\boldsymbol{\theta}_D}(\mathbf{x}), y)] > \mathbb{E}_{(\mathbf{x},y) \sim P_{\mathcal{T}}}[\ell(\phi_{\boldsymbol{\theta}_{D'}}(\mathbf{x}), y)], \tag{1}$$

where $P_{\mathcal{T}}$ denotes the test distribution, $\mathbf{x}$ is a test sample, $y$ is its label, $\ell$ is the loss function (e.g., cross-entropy loss), and $\boldsymbol{\theta}_D$ and $\boldsymbol{\theta}_{D'}$ are parameters of network $\phi$ trained on $D$ and $D'$, respectively.

Notably, directly extending these methods to unlabeled data is infeasible. Existing dataset distillation methods focus on labeled data, whereas knowledge distillation methods, as reviewed in Sec. 2, significantly differ from ours. A detailed comparison is presented in Fig. 2. To address these limitations, we propose a collaborative framework that leverages distributed computation and various task-agnostic models for unlabeled data. Specifically, inspired by (Sun et al., 2024), which shows that using task-agnostic models for target assignment can expedite training, we further enhance efficiency by splitting the data and then parallelly optimizing each split. After optimization, the optimal subset are aggregated to reconstruct a fully optimized dataset, achieving computational efficiency. Formally, we define data optimization with a prior model $\psi$ in each participant in Def. 1.

---

**Definition 1 (Data optimization with prior model $\psi$).** *Given samples $D_X = \{\mathbf{x}_i\}_{i=1}^N$ and a prior model $\psi : \mathbb{R}^m \to \mathbb{R}^l$, data optimization assigns optimal targets $D_Y = \{\mathbf{y}_i\}_{i=1}^N$ for the samples to create $D' = \{\mathbf{x}_i, \mathbf{y}_i\}_{i=1}^N$. We assigns a target $\mathbf{y}_i$ for $\mathbf{x}_i$ as:*

$$D' = \{(\mathbf{x}_i, \mathbf{y}_i) \mid \mathbf{y}_i = \mathbf{W}\psi(\mathbf{x}_i), \forall \mathbf{x}_i \in D_X\}, \tag{2}$$

*where $\mathbf{y}_i$ is the optimized target, and $\psi(\mathbf{x}_i)$ represents the target of $\mathbf{x}_i$, which means the feature representation. $\mathbf{W} : \mathbb{R}^l \to \mathbb{R}^n$ denotes a matrix designed to transform the feature vector $\psi(\mathbf{x}_i)$ from dimension $l$ to $n$ without loss of information (Matoušek, 2008). This transformation aligns the output dimension[a] with that required by the model trained on optimized data $\phi_{\boldsymbol{\theta}_{D'}} : \mathbb{R}^m \to \mathbb{R}^n$.*

---

[a]Here, $n$ is the target dimensionality of $\phi_{\boldsymbol{\theta}_{D'}}$. In practice, each participant produces targets of varying dimensions due to using different prior models. Therefore, to train the model $\phi_{\boldsymbol{\theta}_{D'}}$ on the optimized data $D'$, we employ a random matrix $\mathbf{W}$ to transform all target vectors to a common dimensionality.

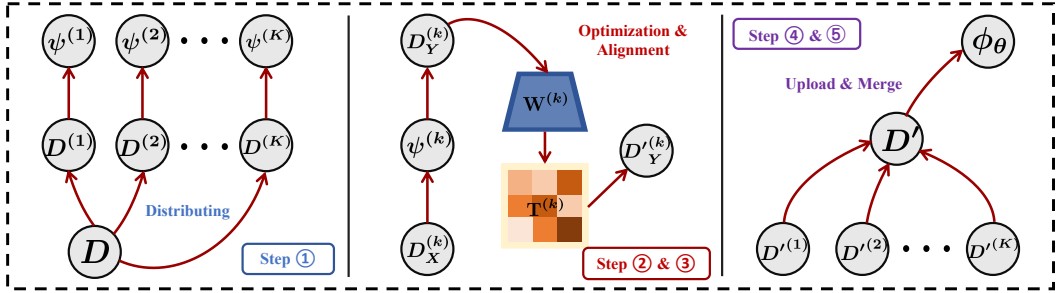

Figure 3: **Lifecycle of the proposed collaborative data optimization framework COOPT**. The framework encompasses an open data platform and multiple participants, involving five key data operations.

## 3.2 OVERVIEW OF THE PROPOSED FRAMEWORK COOPT

COOPT is a collaborative and parallelized framework that includes an open data platform and $K$ participants, each equipped with a distinct prior model. COOPT operates through the five steps:

**Step ①: Data Distributing.** The open data platform initiates the process by randomly partitioning the entire set of unlabeled data $D$ into $K$ non-overlapping subsets. Each participant then downloads one of these subsets from the platform, denoted as $D^{(k)}$, where $k$ denotes the $k$-th participant.

**Step ②: Data Optimization.** Each participant optimizes their respective unlabeled dataset $D^{(k)}$ using the prior model $\psi^k$. This data optimization process, defined in Prop. 1, yields optimized targets $D_Y^{(k)}$ and optimized data $D'^{(k)}$.

**Step ③: Data Alignment.** The heterogeneity of prior models among participants induces significant variations in their distribution of the targets. This issue, referred to as *target distribution inconsistency* (defined in Sec. 3.3), necessitates an alignment strategy (detailed in Sec. 3.4) to align the target distribution across participants. Crucially, each participant needs to align their targets distribution to the most optimal prior model using a learnable transformation matrix $\mathbf{T}^{(k)}$, yielding the final optimized dataset $D'^{(k)}$. Further details are provided in Sec. 3.4.

**Step ④: Data Uploading.** After optimization and alignment, participants upload their optimized datasets $\{D'^{(k)}\}_{k=1}^K$ back to the open data platform.

**Step ⑤: Data Merging.** The platform aggregates all the optimized datasets received from the participants to form a consolidated dataset.

The proposed COOPT enables participants to independently and parallelly optimize their subsets while ensuring consistency through the proposed alignment strategies. Consequently, this approach markedly reduces individual data optimization costs and enhances efficiency.

## 3.3 AN INHERENT CHALLENGE: TARGET DISTRIBUTION INCONSISTENCY

In our collaborative framework, each participant may employ a distinct prior model, leading to inconsistencies in the target distributions, as illustrated in Fig. 6a. For example, participant 1 uses ResNet-18 for optimization, resulting in a target dimension of 512, while participant 2 utilizes ResNet-50, yielding a target dimension of 2,048. Such inconsistencies can negatively impact the generalization capabilities of models trained on the optimized data, as they prevent the models from learning representations that are uniformly representative of the overall data distribution.

## 3.4 AN EFFECTIVE STRATEGY: TARGET ALIGNMENT

To address this issue, a potential solution is to align the target distributions of all participants' prior models with that of the prior model producing the most optimal target distribution, referred to as the *best prior model*. Such alignment can be achieved by utilizing an optimizable transformation matrix to map each participant's target distribution to that of the best prior model (Sun et al., 2024). This alignment strategy ensures consistency across all optimized target distributions.

In summary, *it is crucial to first effectively assess each participant's prior model quality and subsequently train the transformation matrix for alignment.*

**A Metric to Quantify Prior Model Quality.**    Inspired by Wang & Isola (2020), which proposes an optimizable metric a.k.a. *uniform value loss* to achieve feature uniformity on the hypersphere during training, we employ this metric to evaluate the quality of prior models. Notably, (Wang & Isola, 2020) also provide *theoretical validation* of the connection between uniformity and feature quality. Specifically, each participant downloads a small shared dataset $S_X$ from the platform and computes the uniformity value of their prior model on $S_X$. They then upload this value to the platform, enabling it to determine which participant possesses the best prior model. The uniform value is computed as:

$$\mathcal{V}_{\text{uniform}}(\boldsymbol{\psi}; S) \triangleq \log \mathbb{E}_{\mathbf{x}_i, \mathbf{x}_j \sim S} \left[ e^{\tau \|\boldsymbol{\psi}(\mathbf{x}_i) - \boldsymbol{\psi}(\mathbf{x}_j)\|_2^2} \right], \tag{3}$$

where $\boldsymbol{\psi}$ is the prior model, $\tau$ is a hyper-parameter set as 2, consistent with (Wang & Isola, 2020).

A lower uniform value indicates a higher-quality prior model, which optimizes targets of superior quality. Extensive experiments in Fig. 5c demonstrate a strong correlation between this metric and the performance of prior, thereby effectively assessing the quality of targets.

**Alignment.**    Upon identifying the best prior model, denoted as $\boldsymbol{\psi}^\star$, all participants, excluding the best prior model itself, proceed to train an optimizable transformation matrix. Specifically, the participant owning $\boldsymbol{\psi}^\star$ computes its optimized targets, denoted as $\mathbf{S}_{Y^\star}$, on the shared dataset $S_X$. $\mathbf{S}_{Y^\star}$ are then uploaded to the platform, which ensures they are publicly accessible to all participants. Following this, each participant $k$ optimizes a lightweight transformation matrix, denoted as $\mathbf{T}^{(k)}$, on the shared dataset $S_X$. The optimization problem is defined as follows:

$$\mathbf{T}^{(k)} = \arg\min_{\mathbf{T} \in \mathbb{R}^{n \times n}} \{\|\mathbf{T} \cdot \boldsymbol{\psi}^{(k)}(\mathbf{S}_X) - \mathbf{S}_{Y^\star}\|_2^2\}, \tag{4}$$

where $\mathbf{S}_X$ represents the matrix form of $S_X$, suitable for input into the network $\boldsymbol{\psi}^{(k)}$, and $\mathbf{S}_{Y^\star}$ also represents the matrix form of $S_{Y^\star}$. After obtaining the transformation matrix $\mathbf{T}^{(k)}$, the participant can convert the optimized targets for its own data using this matrix: $D_Y{}^{(k)} = \mathbf{T}^{(k)} \cdot \boldsymbol{\psi}^{(k)}(\mathbf{D}_X{}^{(k)})$, where $\mathbf{D}_X{}^{(k)}$ denotes the participant's subset, and $D_Y{}^{(k)}$ are the adjusted targets aligned with the best prior model's target distribution. As illustrated in Fig. 6b, the proposed alignment strategy effectively mitigates target distribution inconsistency.

**Remark on Privacy.**    Notably, this work focuses on optimizing large-scale open-source unlabeled data obtained from publicly available sources, such as the internet. The information transmitted between the platform and participants is targets generated by prior models, which ensures that no direct privacy-sensitive information is exposed. Nevertheless, enhancing mechanisms for robust privacy protection remains a central objective for our future research.

**Remark on Theoretical Effectiveness.**    CoOpt builds upon well-established theoretical foundations. Specifically, RELA Sun et al. (2024) has theoretically demonstrated that leveraging task-agnostic models, such as pre-trained models, can accelerate learning. Notably, CoOpt diverges from RELA in both its objectives and methodology. Specifically, CoOpt introduces a collaborative optimization approach tailored for unlabeled data, targeting a fundamentally distinct problem domain. The challenges associated with collaborative data optimization are unique to CoOpt and remain unaddressed by RELA. Furthermore, the metric we employ to evaluate the quality of prior models, uniform value loss (Wang & Isola, 2020), has been theoretically validated for its effectiveness.

## 4    EXPERIMENTS

We conduct extensive experiments to demonstrate the key advantages of CoOpt in Sec. 4.2. Specifically, *first*, to evaluate its efficacy and efficiency in utilizing unlabeled data, we compare CoOpt with state-of-the-art self-supervised learning methods. *Second*, to examine the necessity of distributed optimization, we further compare CoOpt with centralized optimization approaches. *Third*, we train diverse model architectures on the optimized data, thereby assessing its generalizability and reusability. *Forth*, we demonstrate the potential of CoOpt for continuous data optimization, showing how it continuously enhances data quality. *Furthermore*, we explore factors that influence its effectiveness (Sec. 4.3), including different prior datasets and prior models. *Finally*, we present comprehensive ablation studies (Sec. 4.4) to verify the impact of each module in CoOpt and its lightweight design.

### 4.1    EXPERIMENTAL SETUP

**Datasets and Networks:**    We conduct experiments on both large-scale and small-scale datasets, including ImageNet-1k ($224 \times 224$) (Deng et al., 2009), Tiny-ImageNet ($64 \times 64$) (Le & Yang,

Table 1: **Comparison of CoOpt with Self-Supervised Learning Methods in Accuracy (%) and Training Time (s).** We use four datasets: CF-10 (CIFAR-10), CF-100 (CIFAR-100), T-IN (Tiny-ImageNet), and IN-1K (ImageNet-1K). The best results are marked in **bold**. ↑ means the *performance* improvement over the underlined second-best result. × denotes the factor of *training speed* compared to the second-best result.

| Dataset | Metric | BYOL | DINO | MoCo | SimCLR | SimSiam | SwAV | DCL | CoOpt (Ours) |
|---------|--------|------|------|------|--------|---------|------|-----|--------------|
| CF-10 | Acc. (%) | $82.8 \pm 0.1$ | $82.6 \pm 0.0$ | $82.9 \pm 0.1$ | $83.1 \pm 0.0$ | $79.0 \pm 0.0$ | $82.9 \pm 0.1$ | $\underline{83.9 \pm 0.1}$ | $\mathbf{89.5 \pm 0.1}$ (↑ **5.6**) |
| | Time (s) | 1,376.56 | 1,457.22 | 1,349.56 | 1,114.81 | 1,090.79 | 1,012.74 | 1,783.34 | **540.43** (× **1.87**) |
| CF-100 | Acc. (%) | $51.7 \pm 0.1$ | $51.0 \pm 0.0$ | $57.8 \pm 0.1$ | $55.4 \pm 0.0$ | $44.6 \pm 0.1$ | $53.2 \pm 0.1$ | $\underline{58.2 \pm 0.2}$ | $\mathbf{67.3 \pm 0.1}$ (↑ **9.1**) |
| | Time (s) | 1,406.17 | 1,419.69 | 1,425.80 | 1,103.45 | 1,139.14 | 1,072.44 | 1,701.49 | **548.11** (× **1.95**) |
| T-IN | Acc. (%) | $43.9 \pm 0.2$ | $36.1 \pm 0.0$ | $42.4 \pm 0.2$ | $41.5 \pm 0.1$ | $40.8 \pm 0.0$ | $39.9 \pm 0.1$ | $\underline{44.6 \pm 0.0}$ | $\mathbf{60.3 \pm 0.1}$ (↑ **15.7**) |
| | Time (s) | 7,086.62 | 7,030.90 | 7,133.98 | 5,621.33 | 5,531.92 | 5,540.96 | 9,201.51 | **2,852.67** (× **1.94**) |
| IN-1K | Acc.(%) | $\underline{61.9 \pm 0.1}$ | $52.2 \pm 0.0$ | $57.6 \pm 0.0$ | $58.0 \pm 0.0$ | $55.8 \pm 0.1$ | $57.2 \pm 0.1$ | $60.6 \pm 0.1$ | $\mathbf{69.8 \pm 0.1}$ (↑ **7.9**) |
| | Time (s) | 133,766.19 | 133,156.88 | 150,420.36 | 99,176.29 | 98,656.57 | 96,134.98 | 102,450.84 | **80,096.43** (× **1.20**) |

2015), CIFAR-100 (Krizhevsky et al., 2009a) and CIFAR-10 ($32 \times 32$) (Krizhevsky et al., 2009b). Following previous self-supervised studies (He et al., 2020; Chen et al., 2020a; Grill et al., 2020; Chen & He, 2021; Assran et al., 2023; Zhang et al., 2024), we employ a range of backbone architectures to evaluate the generalizability of our method, including ResNet-{18, 50, 101} (He et al., 2016), ViT (Dosovitskiy et al., 2020), and a series of CLIP-based models (Radford et al., 2021).

**Baselines:** For the unlabeled data, following a widely used benchmark (Da Costa et al., 2022), we compare against state-of-the-art self-supervised (SSL) methods, including: SimCLR (Chen et al., 2020a), BYOL (Grill et al., 2020), DINO (Caron et al., 2021), MoCo (He et al., 2020), SimSiam (Chen & He, 2021), SwAV (Caron et al., 2020), and DCL (Yeh et al., 2022). Notably, we do not compare with knowledge distillation (KD) or dataset distillation (DD) methods, since the training paradigm of KD differs significantly from ours, while DD primarily focuses on the labeled data.

**Evaluation and Metrics:** Following previous benchmarks (Grill et al., 2020; Chen & He, 2021), we evaluate the representation quality of models by evaluating their test accuracy (%) using an offline linear probing strategy. Additionally, computational efficiency quantified by time cost (s).

**Implementation Details:** The proposed algorithm, CoOpt, involves an open data platform and facilitates interaction among multiple participants. The implementation follows five key steps (detailed in Sec. 3.2), and more details are provided in App. C.

1. For step ①: the training dataset is evenly distributed among all participants.
2. For steps ② and ③, in practical applications, *each participant can use publicly pre-trained models or their own models directly as the prior model.* To simulate the diversity of prior models in practical applications, we use a series of pre-trained CLIP-based models.
3. Steps ④ and ⑤ involve uploading data to the open platform for aggregation. Subsequently, for training on optimized data, we use the AdamW optimizer, the same as baselines. The size of mini-batch is set as 128, except for ImageNet-1K, where a mini-batch size of 256 is utilized.

All experiments are conducted using 4 NVIDIA RTX 4090 GPUs. For all experiments, we utilize 3 random seeds and report both the mean and variance of the results. For fair comparisons, all methods in the experiments are executed with the same hyperparameters.

### 4.2 What are the advantages of CoOpt?

**Comparison with SSL Methods.** As shown in Tab. 1, *our CoOpt demonstrates superior performance and efficiency compared to existing self-supervised learning methods.* We also visualize the training dynamic in Fig. 5a. Specifically, CoOpt achieves an improvement of 7.9% over the leading self-supervised approach BYOL on ImageNet-1K. For Efficiency, CoOpt demonstrates a substantial improvememt across various datasets. Notably, on the Tiny-ImageNet, CoOpt achieves a training speed that surpasses the efficient method SimSiam by a factor of approximately ×1.94.

**Comparison with Centralized Optimization.** A key advantage of CoOpt lies in its ability to use diverse prior models in parallel, thereby enabling efficient optimization. To validate this, we compare CoOpt with centralized optimization, where a single model is used to optimize all unlabeled data. Specifically, we consider 10 prior

Table 2: Comparison with Centralized Optimization.

| Method | Time (s) | Acc. (%) |
|--------|----------|----------|
| Centralized | 23.71 | $62.1 \pm 0.1$ |
| Ours | **16.31** | $\mathbf{65.8 \pm 0.1}$ |

Table 4: **Comparison of CoOpt with BYOL Across Diverse Prior Datasets.** For instance, "CIFAR-10 (P)" indicates participants' prior models are trained on CIFAR-10. **Bold** means the best results. Underline indicates the results when the prior dataset is identical to the training data. All models are based on ResNet-18.

| Dataset | BYOL (Baseline) | Our CoOpt (Diverse Prior Datasets) | | | |
|---|---|---|---|---|---|
| | | CIFAR-10 (P) | CIFAR-100 (P) | Tiny-ImageNet (P) | ImageNet-1K (P) |
| CIFAR-10 | $82.8 \pm 0.1$ | $86.6 \pm 0.0$ ($\uparrow$ 3.8) | $80.9 \pm 0.0$ ($\downarrow$ 1.9) | $81.6 \pm 0.1$ ($\downarrow$ 1.2) | **$88.1 \pm 0.0$ ($\uparrow$ 5.3)** |
| CIFAR-100 | $51.7 \pm 0.1$ | $54.9 \pm 0.1$ ($\uparrow$ 3.2) | $60.0 \pm 0.1$ ($\uparrow$ 8.3) | $56.8 \pm 0.0$ ($\uparrow$ 5.1) | **$63.7 \pm 0.0$ ($\uparrow$ 12.0)** |
| Tiny-ImageNet | $43.9 \pm 0.2$ | $38.3 \pm 0.0$ ($\downarrow$ 5.6) | $40.2 \pm 0.1$ ($\downarrow$ 3.7) | $49.0 \pm 0.0$ ($\uparrow$ 5.1) | **$55.8 \pm 0.1$ ($\uparrow$ 11.9)** |
| ImageNet-1K | $61.9 \pm 0.1$ | $31.7 \pm 0.1$ ($\downarrow$ 30.2) | $31.8 \pm 0.0$ ($\downarrow$ 30.1) | $40.5 \pm 0.0$ ($\downarrow$ 21.4) | **$71.2 \pm 0.0$ ($\uparrow$ 9.3)** |

models of varying quality on CIFAR-100. For the centralized setting, we report the mean performance of these 10 independently selected models to ensure fairness. As summarized in Tab. 2, CoOpt *consistently demonstrates superior efficiency and efficacy*, verifying the benefits of distributed over centralized optimization. This improvement arises from the proposed target alignment strategy (Sec. 3.4), which leverages high-quality priors to enhance the target distribution of weaker models.

While one might envision an ideal centralized solution that exclusively employs the best prior model, such an approach is rarely practical in real-world scenarios A major concern is fairness and computational burden, as concentrating all computation on a single party imposes excessive cost and discourages participation. Another challenge is privacy, since high-performing models are typically proprietary, and centralizing them may violate ownership or data-sharing constraints. In contrast, CoOpt collaboratively leverage diverse prior models, achieving competitive performance while substantially reducing the burden on any individual participant.

**Generalizability and Reusability of Optimized Data.** Another key advantage of our optimized data lies in its strong generalizability and reusability: once constructed, it can be directly employed for downstream diverse model training without further modification. To evaluate this advantage, we conduct experiments by training a variety of neural architectures on the optimized data and compare the results to a strong baseline, BYOL, which relies on training from scratch on the original unlabeled dataset. The results are summarized in Tab. 3.

Table 3: Comparison of CoOpt with BYOL on Diverse Networks.

| | Method | |
|---|---|---|
| Network | BYOL | CoOpt |
| ResNet-50 | $60.4 \pm 0.1$ | **$63.8 \pm 0.0$** |
| ResNet-101 | $61.5 \pm 0.2$ | **$65.7 \pm 0.2$** |
| MobileNet-v2 | $24.0 \pm 0.5$ | **$58.1 \pm 0.0$** |
| Efficientnet-b0 | $2.3 \pm 0.2$ | **$70.7 \pm 0.2$** |
| ViT | $38.5 \pm 0.1$ | **$57.8 \pm 0.1$** |

Obviously, CoOpt consistently delivers *significant performance improvements over BYOL across multiple architectures*. In particular, BYOL suffers substantial degradation when applied to lightweight networks such as MobileNet-v2. A plausible explanation is that these models are more sensitive to unstable batch normalization (BN) statistics in early network layers. Instead, our optimized data exhibits *strong generalization* across diverse architectures.

**Continuous Data Optimization.** We further explore a practical scenario where the prior models undergo temporal evolution. For example, a participant's initial model, such as ResNet-50, might be upgraded to a higher-capacity model like ResNet-101 as their computational resources improve. Consequently, the process can be treated as a dynamic and continuous procedure. The detailed description of the process is provided in App. C.4.

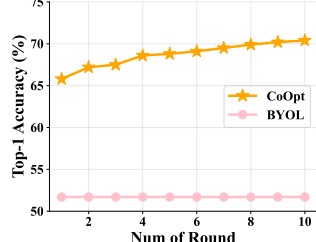

Figure 4: A Practical Scenario: Continuous Optimization.

We simulate this scenario by making random 20% of the participants increase their model capacity in each round. The training curves across 10 rounds on CIFAR-100 are shown in Fig. 4. The results demonstrate that *in CoOpt, as the prior models evolve, the quality of the targets improves, thereby facilitating continuous optimization.* In particular, over 10 rounds, the continuous optimization setting yields a 4.6% performance gain.

### 4.3 WHAT INFLUENCE THE EFFECTIVENESS OF CoOpt?

We investigate the key factors that influence the effectiveness of the optimized data. Since the optimization of unlabeled data in CoOpt relies solely on task-agnostic prior models, we focus on two primary aspects in these experiments: prior datasets and prior models.

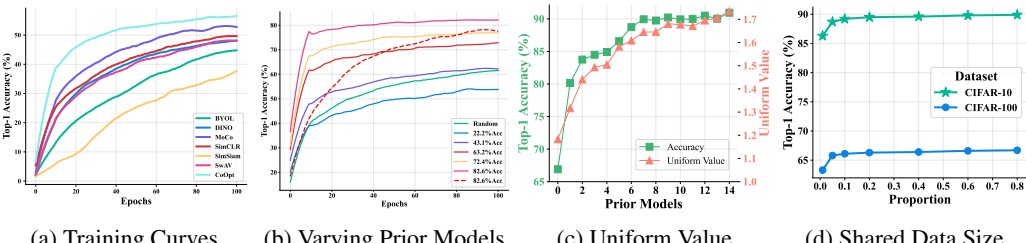

(a) Training Curves     (b) Varying Prior Models     (c) Uniform Value     (d) Shared Data Size

Figure 5: **Comprehensive Analysis of CoOpt.** *(a) Training curves:* Comparison of SSL methods and our CoOpt. *(b) Prior Models With Varying Accuracies.* Even with a very weak prior model, CoOpt accelerates the early-stage training. *(c) Correlation Verification:* Verify the correlation between the uniform value and performance. *(d) Influence of shared data size:* As shared data's size increases, the performance gains diminish.

**Influence of Prior Datasets.** To rigorously evaluate the influence of prior datasets that are used for training prior models, we perform an analysis across scenarios where the prior datasets used for prior models either align with or differ from the unlabeled training dataset. For instance, in the aligned scenario, the training dataset is CIFAR-10, and the prior models are also trained on CIFAR-10 (CIFAR-10 (P)). Conversely, in the divergent scenario, the training dataset remains CIFAR-10, while the prior models are trained on CIFAR-100 (P). Details of these models are provided in App. C.3. We evaluate our approach on four publicly available datasets, with the results summarized in Tab. 4.

Obviously, for *all* unlabeled training datasets, *employing prior models trained on ImageNet-1K consistently yields notable performance gains,* attributed to their strong generalization abilities. This observation is particularly relevant in practical applications, as most publicly available pre-trained models are derived from ImageNet-1K or even larger datasets. On the other hand, for complex training datasets, leveraging prior models trained on simpler datasets may result in degraded performance compared to BYOL. This is likely due to the limited informativeness of simpler prior datasets, which provide weaker guidance. We further examine the influence of weak models in Fig. 5b.

**Special Cases of Prior Models: Human or Weak Involvement.** In real-world applications, extreme cases arise due to the varying capabilities of participants. For example, some participants have extensive resources and can employ human annotators for labeling, while others may have limited resources and rely on weak models with inferior generalization abilities. In this experiment, we define weak models as those trained during intermediate stages that are even far from convergence. To simulate the conditions, in addition to the prior models used in the first experiment, we incorporate 5 prior models, either human or weak models to the data optimization process. The

Table 5: Comparison of CoOpt with BYOL in Presence of Human or Weak Prior Models.

| Method | Prior Models | | Dataset |
| | Human | Weak | CIFAR-10 |
| --- | --- | --- | --- |
| BYOL | – | – | 82.8 ± 0.1 |
| CoOpt | ✗ | ✗ | 89.5 ± 0.1 |
| | ✗ | ✓ | 89.2 ± 0.2 |
| | ✓ | ✗ | **90.5 ± 0.1** |
| | ✓ | ✓ | 89.8 ± 0.1 |

results are summarized in Tab. 5. Surprisingly, even the inclusion of weaker models contributes to enhancing the final performance, indicating that such models can still provide valuable information. Moreover, it is important to note that *the integration of high-capacity, human-like models results in significant performance improvements.*

**Extreme Cases of Prior Models: All are Weak.** Moreover, we conduct experiments with only weak model, as shown in Fig. 5b. Here, The dashed line represents the training curve of BYOL (baseline), while the solid lines correspond to prior models with different accuracies. "Prior model (BYOL)" indicates the use of BYOL as the prior model, and the accuracies of the other prior models are all lower than that of BYOL. While a stronger prior model does yield better performance, our results demonstrate that *even with a moderately weak prior model with approximately 75% accuracy, our method can still outperform the baseline.* More importantly, even when prior models are substantially weak, CoOpt *still significantly outperform baseline in the* **early training stages.**

### 4.4 ABLATION STUDY

**Can Uniform Value Effectively Assess the Quality of Prior Models?** To evaluate the effectiveness of uniform value in estimating the quality of prior models, we employ a diverse set of prior models and compute both their uniform values and corresponding test accuracies. To quantify the relationship

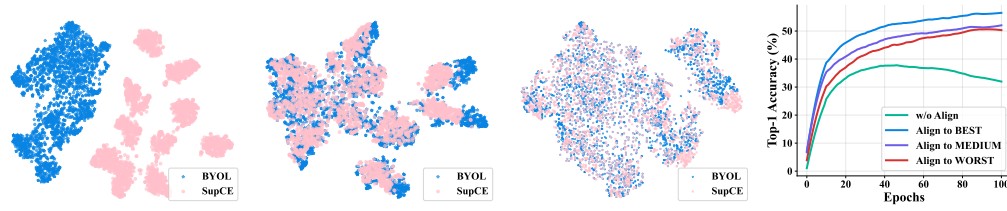

(a) Before Alignment    (b) Align (BYOL → CE)    (c) Align (CE → BYOL)    (d) Training curves.

Figure 6: **Effectiveness of Target Distribution Alignment.** (a), (b), (c): Visualization of t-SNE for optimized targets generated by two distinct models (BYOL (acc. = 82%) and SupCE (acc. = 90%).) Aligning to the worse model (c) results in diminished target quality. (d): Training curves with and without alignment.

between these two measures, we adopt the Spearman rank correlation coefficient[2] $\rho$ (Zar, 2014) to quantify the association between Uniform Value and accuracy. As illustrated in Fig. 5c, the Spearman rank correlation coefficient is $\rho = -0.9714$, *indicating a strong correlation between uniform value and model performance, thereby effectively assessing the quality of prior models.*

**Why Target Distribution Alignment is Necessary?**    In real-world applications, participants often adopt diverse prior models, leading to inconsistencies in the target space (Fig. 6a). To verify the necessity of alignment, we conduct an ablation study with and without alignment, and the training curves in Fig. 6d show that our alignment method improves performance by 16.9%, veifying its effectiveness. Furthermore, We compare three strategies: aligning to the best prior model (ours), a medium-quality prior model (purple line), and the worst prior model (red line). All strategies outperform the unaligned baseline, but aligning to the best prior model yields the largest performance gains, as reported in Tab. 7 of App. C.5. To analyze the underlying reasons, we employ t-SNE visualization. As shown in Fig. 6b and Fig. 6c, alignment with a high-quality model enhances the representational capability of the targets, whereas alignment with a poor model diminishes it.

**How Much Shared Unlabeled Data Is Enough?**    The shared unlabeled data $S$ is used to estimate the uniform value and compute the transformation matrix for target alignment, as detailed in Sec. 3.4. To explore the influence of the data size, we vary the proportion of shared data on CIFAR-10 and CIFAR-100, as shown in Fig. 5d. Obviously, *as the size of the shared data increases, performance gains become marginal*, indicating that a very small fraction (around 0.05%) is sufficient for accurate estimation and alignment. We further validate this on ImageNet-1K (Tab. 8 in App. C.6), where only 0.001% of the data achieves comparable results to larger proportions. This minimal requirement imposes negligible additional overhead, ensuring scalability to very large datasets.

**Does Alignment Incur Significant Overhead?**    We report the computational cost of uniform value estimation and alignment in Tab. 6, verifying that *both incur only negligible cost*. This efficiency stems from the former requiring just a single forward pass to obtain targets, while the latter involves optimizing a lightweight matrix.

Table 6: Comparison of Time (s) on ImageNet-1K.

| BYOL | Uniform value | Alignment |
|---|---|---|
| 133,766.19 | 139.16 | 36.97 |

## 5 CONCLUSION

We introduce CoOPT, a pioneering data-centric, parallelized, and efficient framework for collaborative optimization of unlabeled data. This data-centric approach results in architecture-agnostic optimized data that are reusable across diverse network architectures, while simultaneously reducing the number of training iterations required, thereby enhancing overall efficiency. Furthemore, within CoOPT, we identify a critical issue: Target Distribution Inconsistency, which arises from the diversity of prior models used in data optimization. To mitigate this, we propose a lightweight target alignment strategy. Extensive experiments demonstrate the superior effectiveness and efficiency of the CoOPT framework across diverse datasets and architectures. One limitation is that when all prior models are extremely weak, the overall performance inevitably degrades. As future work, we aim to develop advanced strategies to more effectively exploit optimized data derived from all extremely weak priors, with our current study verifying efficiency in the early training stage.

---

[2]The formula is: $\rho = 1 - \frac{6 \sum d_i^2}{n(n^2-1)}$, where $d_i$ represents the difference between the ranks of each pair of observations and $n$ denotes the number of observations.

ETHICS STATEMENT

This work has been conducted *in accordance with the ICLR Code of Ethics* and upholds the principles of responsible and transparent research. Our study does not involve human participants, personal or sensitive data, or any elements necessitating institutional ethics board approval. All datasets employed are publicly available and accompanied by licenses, with full acknowledgment and attribution provided to the respective creators. To foster openness and reproducibility, we release our implementation code together with experimental configurations to enable verification and extension by the research community. We further confirm that no conflicts of interest or external sponsorships have influenced the conception, design, execution, or reporting of this work.

REPRODUCIBILITY STATEMENT

Importantly, *complete code of our method is provided in the supplementary materials.* Moreover, detailed descriptions of the datasets, model architectures, optimization configurations, and training protocols can be found in Sec. 4.1 of the main paper as well as in App. C. Together, these resources enables researchers to reproduce the results in a reliable and transparent manner.

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

## A   THE USE OF LARGE LANGUAGE MODELS (LLMs)

In this work, Large Language Models (LLMs) are used at the sentence level to support linguistic refinement. Their role was limited to enhancing the grammar and readability of the manuscript. All research ideas, methodological designs, experimental procedures, and analytical conclusions are entirely original and the sole work of the authors.

## B   RELATED WORK

**Self-supervised Learning: A Model-Centric Perspective.**   Self-supervised learning (Chen et al., 2020b) aims to exploit the intrinsic relationships within unlabeled data. For example, InstDisc (Wu et al., 2018) uses instance discrimination as a pretext task. Building on this, CMC (Tian et al., 2020) proposes to use multiple views of an image as positive samples and take another one as the negative. MoCo (He et al., 2020) significantly increases the number of negative samples but uses a simplistic strategy for selecting positive samples. SimCLR (Chen et al., 2020a) highlights the importance of hard positive sample strategies by introducing data augmentation. Notably, BYOL (Grill et al., 2020) discards negative sampling and surpasses the performance of SimCLR (Chen et al., 2020a).

**Summary.**   *They are constrained to specific architectures and incur high computational costs due to the reliance on large batch sizes or memory banks.*

**Knowledge Distillation: Optimzing Targets.**   Knowledge distillation (Hinton, 2015) employs soft labels generated by teacher models to improve the performance of a student model and expedite its training (Dong et al., 2023). Many following works aim to enhance the use of soft labels for more effective knowledge transfer. For example, WSLD (Zhou et al., 2021) analyzes soft labels and distributes different weights for them from a perspective of bias-variance trade-off. DKD (Zhao et al., 2022) decouples the logits and assigns different weights for the target and non-target classes. Moreover, several studies (Yim et al., 2017; Dong et al., 2023) demonstrate that knowledge distillation can accelerate the training process.

**Dataset Distillation: Optimizing Both Samples and Targets.**   Dataset distillation (Wang et al., 2018) aims to learn a compact distilled dataset that can achieve comparable performance to the original dataset with less training cost. The majority of methods focus on optimizing images, which can be categorized into five primary approaches (Lei & Tao, 2023): meta-learning frameworks (Wang et al., 2018; Zhou et al., 2022), gradient matching (Zhao et al., 2020; Zhao & Bilen, 2021), distribution matching (Zhao & Bilen, 2023; Yin et al., 2023), trajectory matching (Cazenavette et al., 2022; Guo et al., 2024), and decoupling frameworks (Yin et al., 2023; Sun et al., 2024). Recently, some studies (Shang et al., 2025; Qin et al., 2024) have shifted focus towards label distillation, aiming to obtain high-quality soft labels. This approach has demonstrated notable efficiency and effectiveness.

## C   EXPERIMENTAL DETAILS

### C.1   IMPLEMENTATION DETAILS

**Hardware Setup.**   All experiments are conducted using 4 NVIDIA RTX 4090 GPUs. For fair comparisons, all methods in the experiments are executed with the same hyperparameters.

**Unlabeled Training Dataset Split.**   In our framework, for all experiments, the unlabeled dataset is evenly distributed among all participants.

### C.2   DIVERSE PRE-TRAINED PRIOR MODELS.

We utilize 10 CLIP-based pre-trained prior models varying from small-scale to large-scale architectures, which are downloaded using the torchvision package. Unlabeled data is equally split among participants.

## C.3 DIVERSE PRIOR DATASETS

In this set of experiments, we investigate the effectiveness of our proposed COOPT when the prior datasets are either similar to or distinct from the training datasets. For each prior dataset, we train 4 different models to serve as prior models. Specifically: These 4 models are trained using two paradigms (supervised and unsupervised learning) and two architectures (ResNet-18 and ViT). For each training dataset, the data is evenly distributed among participants. Each prior model is assigned 1/4 of the unlabeled dataset, which it optimizes independently. The results are then aggregated on the open platform.

## C.4 CONTINUOUS DATA OPTIMIZATION

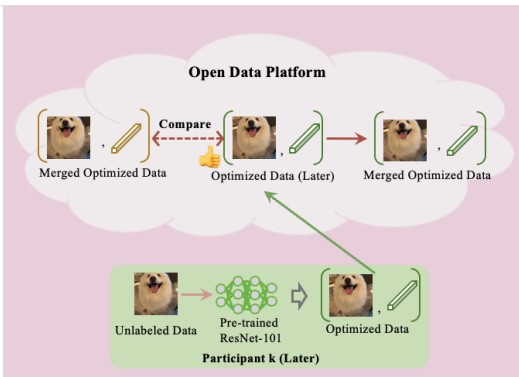

Figure 7: **Common Target Conflict Scenarios**

In real-world scenarios, model architectures and computational resources often evolve over time, necessitating a dynamic and continuous approach to data optimization. Unlike static optimization, where data is processed only once, this framework accommodates temporal model evolution and repeatedly refines optimized data to achieve superior results. Below, we describe the core components of this framework.

**Dynamic Model Evolution.**    Initially, all data is optimized using participants' current models. Over time, as computational resources improve, participants upgrade to higher-capacity architectures. These upgraded models exhibit stronger feature extraction capabilities, enabling further refinement of previously optimized data. This dynamic evolution transforms the optimization process into a continuous improvement cycle.

**Open Platform for Optimized Data Comparison.**   A key feature of this framework is the inclusion of an open platform for optimized data submission and evaluation. Participants provide newly optimized data, which is compared against previously optimized data to retain the superior results. This comparison leverages evaluation metrics such as the uniform value of the prior model, ensuring that the dataset evolves toward higher optimization quality over successive iterations.

**Iterative Optimization Process**    The overall process is illustrated in Fig. 7. The framework operates as a loop:

1. Data is initially optimized by participants' models.

2. As models evolve, data is re-optimized to reflect the improved capacities of the upgraded architectures.

3. The open platform compares new and old optimization results, retaining the higher-quality data.

4. This process repeats over multiple interaction rounds, progressively enhancing the dataset.

Table 7: Performance Under Three Selection Scenarios.

| Dataset | BYOL | No align | Best | Medium | Worse |
|---|---|---|---|---|---|
| CIFAR-100 | $51.7 \pm 0.1$ | $44.7 \pm 0.0$ | $\mathbf{65.3 \pm 0.0}$ | $63.1 \pm 0.0$ | $60.1 \pm 0.1$ |

## C.5 VARIOUS ALIGNMENT STRAGETIES

As shown in Fig. 6d, even when the best prior model is mis-selected (i.e., prior models are aligned to a medium-performing prior or even the worst prior, represented by the purple and red lines, respectively), the overall performance still surpasses the "no align" baseline. Furthermore, we present the final performance in Tab. 7. The results suggest that our method continues to outperform the baseline methods under three selection scenarios and demonstrates robustness to prior model mis-selection.

Table 8: Influence of Shared Data Size on Large-Scale ImageNet-1K.

| Dataset | BYOL | 0.001 | 0.1 |
|---|---|---|---|
| ImageNet-1K | $61.9 \pm 0.1$ | $68.8 \pm 0.1$ | $68.7 \pm 0.0$ |

## C.6 SIZE OF SHARED DATA

The shared unlabeled data $S$ is used to estimate the uniform value and compute the transformation matrix for target alignment, as detailed in Sec. 3.4. The shared dataset is public and randomly selected. This eliminates privacy concerns and does not require matching any specific data distribution, thus avoiding issues of distribution mismatch or privacy leakage. Moreover, we further evaluated our method on the large-scale ImageNet-1K. As shown in Tab. 8, only 0.001% of the data samples are sufficient to achieve performance comparable to using larger amounts of data. This minimal requirement imposes negligible additional overhead, ensuring scalability to very large datasets.

