# OpenReview forum: "Beyond Model-Centric: Collaborative Data Optimization for Reusing and Sharing"
_ICLR.cc/2026/Conference — Submitted to ICLR 2026_

### Official Review · Reviewer_PxTb · 2025-10-30

**Soundness:** 3
**Presentation:** 3
**Contribution:** 2
**Rating:** 6
**Confidence:** 3

**Summary:**

The paper addresses the problem of improving training efficiency on large unlabeled datasets and proposes CoOpt, a framework that enables multiple participants to augment distinct subsets of data by assigning soft targets. These targets are further calibrated using a small shared dataset, after which the optimized subsets are uploaded and merged into a reusable dataset. The merged optimized dataset can then be transferred to any downstream architecture.

**Strengths:**

- The proposed method introduces a parallel framework for collaborative data optimization, addressing efficiency and reusability of knowledge implicitly stored in unlabeled data across different model architectures.
- It demonstrates stronger empirical results compared to existing self-supervised learning frameworks.
- The paper provides an in-depth robustness analysis of the proposed method by varying factors such as prior models and shared data size.

**Weaknesses:**

- The paper states that the same hyperparameters are used across all methods in the experiments; however, this may not be fair to competing baselines, which might require task-specific tuning.
- The exploration of scalability (e.g., increasing the number of participants or dataset size) remains limited.
- It remains unclear whether the use of diverse prior models consistently benefits representation learning, particularly in scenarios with imbalanced data distributions across participants.

**Questions:**

- The authors claim that the generalizability of optimized data is a key advantage. However, could it be that BYOL primarily suffers from unstable training, and that with better-tuned parameters, it might exhibit stronger generalization than CoOpt?
- Is it possible that CoOpt’s stability comes at the cost of limited generalization due to its dependence on the prior models—and if so, could such limitations be hard to detect?
- Can the framework be extended to other modalities?
- Does the framework yield stable results under different methods of parameter initialization?

---

### Official Review · Reviewer_U5RS · 2025-10-31

**Soundness:** 2
**Presentation:** 2
**Contribution:** 3
**Rating:** 4
**Confidence:** 4

**Summary:**

This paper proposes COOPT, a data-centric collaborative framework that aims to encode knowledge into data so that optimized unlabeled data can later be consumed by different downstream models. The motivation is timely: current self-supervised learning (SSL) pipelines remain model-centric and expensive, learned knowledge is locked in specific parameters, and multi-party scenarios often have separate models and unlabeled data but lack an effective way to share what has been learned. COOPT addresses this by letting multiple participants use their own prior models to generate targets on disjoint subsets of the unlabeled pool, then aligning these targets to a common space and aggregating them into a reusable optimized dataset. This is an interesting shift from optimizing a single model to optimizing the data itself.

**Strengths:**

- Data-centric reframing: The paper makes a clear conceptual move from a model-centric to a data-centric view. By treating the reusable asset as an optimized dataset rather than a trained model, it decouples the learned supervision from any specific architecture. This is supported by the results on lightweight models (e.g., Table 3), where conventional SSL baselines underperform but models trained on the optimized data still obtain competitive accuracy.

- Tolerance to weak priors: The framework remains effective even when the participating priors are of mixed or relatively low quality. The observation in Fig. 5b that a pool including weaker participants can still outperform a strong single-model SSL baseline suggests that the collaborative alignment step itself adds value, instead of merely relaying the strongest teacher. This is a useful property for realistic multi-party settings.

- Separation of one-time supervision generation from repeatable training: The method separates the expensive stage (running priors on unlabeled data and aligning the targets) from the cheap stage (training arbitrary downstream models on the optimized data). In practice, this leads to training curves that start from a higher point (Fig. 5a), which reduces the time-to-use for later users of the data.

- Can be interpreted as a form of amortized supervision: COOPT can be interpreted as a form of amortized supervision (I like it). A set of priors generates and aligns supervision once and stores it together with the data, so that later learners can consume this supervision without re-running the expensive guidance. This is close in spirit to amortized inference, which pays the cost once and reuses the result, but here the amortized object is target generation across heterogeneous teachers, and an additional cross-teacher alignment step is required.

**Weaknesses:**

## Major

### 1. Implausible claim about alignment sample size
Appendix C.6 states that on ImageNet-1K only 0.001% of data samples are sufficient for alignment. ImageNet-1K has about 1.28M training images, so 0.001% $\times$ 1.28 $\times 10^6 = 12.8$ images. At the same time, the method section defines an alignment step that learns a transformation matrix $T^{(k)}\in\mathbb{R}^{n\times n}$ to map each participant’s targets into the space of the best prior, as in Eq. (4). Under the current description it is difficult to believe that 13 images are enough to stably estimate and generalize such a matrix, even if all features have already been projected to a common dimensionality. The curves in Fig. 5d and the numbers in Table 8 look much more consistent with 0.1% (around 1280 images) than with 0.001%. This suggests either that the reported percentage is a typo or that the actual projection dimensionality $n$ used in experiments was very small but not reported. In either case, the current description is not reproducible, and the statement that 0.001% is sufficient should be clarified or qualified.

### 2. Current results do not match the breadth of the reusable data asset claim
The paper presents COOPT as producing optimized data that can be reused and is independent of the choice of downstream architecture. In the SSL literature, claims of this type are usually supported by evaluations on diverse downstream tasks, including detection and segmentation, because the value of SSL is judged by cross-task transfer rather than by classification alone. In this paper, all evaluations are classification only, mostly linear probing on CIFAR and ImageNet. These results do show that COOPT is an efficient way to obtain stronger classification supervision from unlabeled data. They do not yet show that the optimized data can be reused across task families. To support the full claim, at least one non-classification downstream task is needed. Otherwise, the claim should be narrowed to semantic or discriminative tasks.

### 3. Bias from the choice of priors
Appendix C.2 states that the experiments mainly rely on 10 CLIP-based pretrained prior models. CLIP style contrastive encoders are trained to produce semantic invariance. They tend to preserve category-level content and to downweight spatially precise information that is important for detection and segmentation. If nearly all priors that generate the targets have this inductive bias, then the resulting optimized targets will also be biased toward semantic or discriminative tasks. This is not a defect of the COOPT framework itself. It is a limitation of the current experimental setup. As long as the priors are mostly CLIP-like, the paper should make it clear that the conclusion is strongest for semantic classification style tasks. If the authors believe that COOPT can also support spatially sensitive tasks, it would help to show results with priors that retain spatial detail, such as MAE or segmentation pretrained backbones, and to discuss how alignment behaves in that case.

### 4. Unclear separation between random projection $W$ and learnable transformation $T$
Section 3.1 introduces a random matrix $W$ to map heterogeneous prior outputs into a common dimensionality and even describes this as being done without loss of information. Later, Section 3.4 introduces a learnable matrix $T^{(k)}\in\mathbb{R}^{n\times n}$ to align each participant to the best prior. In the current writing it is not clear why a second alignment step is needed if $W$ already provides a lossless mapping. A reasonable interpretation is that $W$ resolves dimensionality differences and $T$ resolves distributional or style differences. Since target alignment is the core mechanism of COOPT, this two stage design should be made explicit.

## Minor
### 5. Sign error in the uniformity formula
Eq. (3) defines the uniformity score as $\mathcal{V}(\psi; S)=\log \mathbb{E}_{x_i, x_j \sim S}\left[  \exp\left(    \tau \bigl\|\bigl\| \psi(x_i) - \psi(x_j) \bigr\|\bigr\|^2  \right)\right]$, and the text explains that a lower value indicates better representation quality. As written, the expression increases when pairwise distances increase, which is opposite to that explanation. The common form in Wang and Isola (2020) uses a negative sign in the exponent. Since this score is used to select the best prior, a sign error would change which prior is chosen and can affect the whole pipeline. This should be checked.

### 6. Missing discussion on storage and communication overhead
The paper highlights training time efficiency and compares COOPT with standard SSL in terms of computation. At the same time, the framework assumes a central or open platform that stores and serves high-dimensional aligned targets produced by multiple participants. At scales of tens or hundreds of millions of images, storing $N\times d\times 4$ bytes with $N$ images and aligned feature dimension $d$ is not negligible. Uploading these targets from multiple parties is not negligible either. A short discussion of this systems side cost would make the efficiency claim more balanced.

**Questions:**

- Could the authors please clarify or correct the "0.001% of ImageNet" alignment data claim? See Weakness #1.
- Given the reliance on semantic CLIP priors, how can the "reusable asset" claim be justified for spatial tasks like detection, and how would alignment handle heterogeneous priors (e.g., MAE and VAE)? See Weaknesses #2 and #3.
- Could the authors please clarify the necessity and distinct roles of the "lossless" random projection $W$ vs. the learnable alignment $T$? See Weakness #4.
- Could the authors please confirm if the sign in the uniformity formula (Eq. 3) was correctly implemented in the experiments? See Weakness #5.
- Could the authors provide a brief analysis of the storage and communication overhead required by the framework at scale? See Weakness #6.
- If the alignment mechanism (Sec 3.4) maps all participants to the best prior, what specific value do the weaker priors (Fig. 5b, Table 5) actually contribute to the final merged dataset?

---

### Official Review · Reviewer_a263 · 2025-11-01

**Soundness:** 2
**Presentation:** 2
**Contribution:** 2
**Rating:** 4
**Confidence:** 4

**Summary:**

This study explores the idea that uses several off-the-shelf pretrained models to produce embeddings for each image and then maps each embedding to a shared target one to form a compact representation. Then the representation is utilized to train a new model as a new representation learning method. It is able to achieve unlabeled data aggregation and optimization in a parallellized and collaborative manner. It achieves improvement on ImageNet-1K over BYOL to testify its significance.

**Strengths:**

- CoOPT is able to leverage strong pretrained models by using off-the-shelf pretrained models to bootstrap a high-quality "gold" representation space without requiring label information or heavy supervision.
- The enriched "label" for each image reduces the following pretraining burden as this new representation enables new pretraining tasks on different models without new heavy label acquisation processes.
- The method utilizes a uniform metric to quantify prior model quality beforehand , which is practically important and interesting.
- CoOPT generally achieves a better performance over various SSL methods on different network structures and sometimes outperforms the baseline by a large margin.

**Weaknesses:**

- Even though this study claims the difference of CoOPT from knowledge distillation, the pipeline is still considered a two-stage knowledge distillation method.  It can also be interpreted as a knowledge distillation method using the ensembled features of SSL models to teach another models without using any extra labels. The alignment loss as in Eq. 4 can also be interpreted as an extra loss on the projection layer to make the representation more compact. There have been quite a few knowledge distillation works in SSL that fit this idea, e.g. [1, 2] that should be discussed and compared with.
- As CoOPT also lies in the realm of knowledge distillation, the comparison between CoOPT with other pretraining methods such as BYOL and SwAV is then deem less fair. The authors are encouraged to compare with other knowledge distillation works in SSL for a more convisible conclusion.
- There is a fatal typo in Eq. 3 where there should be a negative sign before the exponent. This causes a signifiant confusion when presenting the idea of uniform value as the authors state "a lower uniform value indicates a high-quality prior model".

[1] Joshi, Siddharth, Jiayi Ni, and Baharan Mirzasoleiman. "Dataset Distillation via Knowledge Distillation: Towards Efficient Self-Supervised Pre-Training of Deep Networks." arXiv preprint arXiv:2410.02116 (2024).
[2] Bhat, Prashant, Elahe Arani, and Bahram Zonooz. "Distill on the go: Online knowledge distillation in self-supervised learning." Proceedings of the IEEE/CVF Conference on computer vision and pattern recognition. 2021.

**Questions:**

Despite the better performance of CoOPT over other SSL methods, the reviewer is afraid that CoOPT is still a knowledge distillation method and thus the comparisons in the study are considered less fair.

---

### Meta-Review · Area_Chair_q9iT · 2026-01-03

**Summary:**

This paper presents a method that is, in essence, a reformulation of established knowledge distillation techniques and fails to meet the standards for publication due to fundamental and irreparable shortcomings.

Lack of Novelty and Outdated Scope: The core contribution is not meaningfully differentiated from prior knowledge distillation works. The evaluation is critically outdated, omitting all relevant methods published after 2023, which fundamentally invalidates its claim to state-of-the-art status and demonstrates a lack of engagement with the current field.

Unreliable Claims: The technical presentation is undermined by a critical error in a core equation. Most severely, the paper's central empirical claim—that effective model alignment can be achieved with only ~13 images (0.001% of data)—is mathematically implausible and indicates either a major reporting error or non-reproducible results.

Incomplete Validation and Unfair Comparisons: The authors' broad claim of creating a reusable, task-agnostic data asset is unsupported, as evaluation is confined only to classification tasks using semantically-biased priors (CLIP). Furthermore, the failure to adequately compare against the correct baseline methods (other SSL distillation works) and the lack of response to reviewer critiques render the experimental analysis incomplete and the comparisons unfair.

Overall, the work fails to address core reviewer concerns regarding fairness and recency. It is below the acceptance threshold.

**Reviewer Concerns:**

No response from authors is submitted.

**Reviewer Scores:**

Given no response, no one would change the score.

---

### Decision · Program_Chairs · 2026-01-26

Reject